# Disease-Centric Vision-Language Pretraining with Hybrid Visual Encoding for 3D Computed Tomography

Bowen Shi [1 2 3]   Weiwei Cao [1 2 4]   Ruifeng Yuan [1 2 5]   Wanxing Chang [1 2]   Wenrui Dai [3]   Hongkai Xiong [3]
Ling Zhang [1]   Jianpeng Zhang [1 2 4]

## Abstract

Vision–language pre-training (VLP) holds great promise for general-purpose medical AI by leveraging radiology reports as rich textual supervision, yet existing methods struggle with 3D CT imaging due to inefficient visual backbones and coarse semantic alignment. To address these issues, we propose a tailored VLP framework featuring three key components: (1) a CNN–ViT hybrid encoder that replaces ViT's patch embedding with a 3D CNN backbone to efficiently capture local anatomical details while preserving global attention and compatibility with pre-trained cross-modal priors; (2) a disease-level contrastive learning mechanism using learnable query tokens to dynamically extract disease-specific semantics from full reports and align them with corresponding visual features, thereby disentangling distinct diseases within the same anatomical region; and (3) a diagnosis-aware prompt strategy that employs real clinical phrases and aggregated disease prototypes to bridge the pre-training–inference gap and enhance zero-shot diagnostic reliability. Our model achieves state-of-the-art performance on CT-RATE (84.4% AUC, +5.1%) and Rad-ChestCT (75.4% AUC, +5.4%), with even larger gains (+9.8% AUC) on a challenging 60-disease benchmark, and demonstrates strong transferability to radiology report generation, underscoring the generality and clinical utility of our approach.

---

[1]DAMO Academy, Alibaba Group [2]Hupan Lab, 310023, Hangzhou, China [3]Shanghai Jiao Tong University, China [4]Zhejiang University, China [5]Fudan University, China. Correspondence to: Wenrui Dai <daiwenrui@sjtu.edu.cn>, Jianpeng Zhang <jianpeng.zhang0@gmail.com>.

*Proceedings of the 43$^{rd}$ International Conference on Machine Learning*, Seoul, South Korea. PMLR 306, 2026. Copyright 2026 by the author(s).

## 1. Introduction

Vision–language pre-training (VLP) (Li et al., 2021; Radford et al., 2021; Li et al., 2023; Zhai et al., 2023; Shi et al., 2024) has introduced a novel paradigm for building general-purpose, versatile medical artificial intelligence systems in recent years. Unlike traditional supervised learning that relies on human-defined, limited categorical labels, VLP leverages clinically generated radiology reports as rich, nuanced linguistic supervision signals to jointly model visual and textual information within a unified semantic space. This paradigm captures semantic associations between images and text, endowing models with zero-shot reasoning and cross-task generalization capabilities, thereby supporting diverse clinical applications such as multi-disease diagnosis and report generation.

Despite the promise of medical VLP for general-purpose diagnosis, current methods face two key limitations. First, visual architectures are poorly suited to 3D medical imaging. Most models repurpose backbones originally designed for natural 2D images. ViT-based approaches (Hamamci et al., 2024a; Shui et al., 2025), when naively extended to 3D computed tomography (CT), suffer from excessive computational cost due to volumetric patchification and often sacrifice spatial resolution, thereby losing critical details of small lesions. In contrast, 3D convolutional neural networks (CNNs) (Blankemeier et al., 2026; Cao et al., 2025) better capture local anatomical textures but deviate from mainstream vision–language model (VLM) architectures (Radford et al., 2021; Zhai et al., 2023), limiting their ability to leverage cross-modal priors learned during large-scale pre-training and hindering effective transfer in joint representation learning. Second, semantic alignment remains coarse. Mainstream methods (Hamamci et al., 2024a; Blankemeier et al., 2026) rely on global image–text contrastive learning, ignoring fine-grained correspondences between localized pathological findings and their textual descriptions. While recent work such as fVLM (Shui et al., 2025) aligns representations at the anatomical level, distinct pathologies within the same organ, such as "atelectasis", "emphysema", and "lung nodules", are still grouped under a generic label like "abnormal". This semantic conflation obscures disease-

specific image–language associations, impairing both diagnostic discrimination and model interpretability.

To overcome these limitations, medical VLP stands to benefit significantly from advances in general-purpose foundation models. On one hand, modern VLMs offer rich, cross-modal priors—pre-trained visual-textual semantic alignments—that enable visual representations to better interpret clinical language in radiology reports, provided architectural compatibility is maintained. On the other hand, large language models (LLMs) (OpenAI, 2023; Gemini Team Google, 2023; DeepSeek-AI, 2024; Yang et al., 2025) can automatically extract disease-level labels from unstructured radiology reports, particularly valuable when manual annotations are scarce, thereby enabling fine-grained, disease-centric alignment. *We argue that a co-design strategy integrating expressive, VLM-compatible visual representations with LLM-augmented, disease-specific learning objectives is essential to translate the clinical potential of foundation models into effective medical VLP.*

To this end, we propose a novel VLP framework tailored for 3D CT imaging, centered on the co-design of efficient representation, fine-grained alignment, and robust inference. First, we introduce a CNN–ViT hybrid encoder that replaces the standard ViT patch embedding module with a 3D CNN backbone and integrates multi-scale features. By initializing from a pre-trained ViT (Dosovitskiy et al., 2021), our hybrid encoder preserves global attention mechanisms while enabling effective integration of cross-modal priors; simultaneously, its 3D CNN backbone efficiently captures local anatomical details across multiple scales, yielding expressive representations for 3D medical volumes. Second, we propose a disease-level contrastive learning mechanism that employs learnable query tokens to dynamically extract disease-specific semantics from full radiology reports and explicitly aligns them with corresponding visual features. This fine-grained alignment disentangles co-occurring pathologies within the same anatomical region, representing the finest-grained semantic alignment reported to date. Crucially, the mechanism flexibly leverages both ground-truth and LLM-extracted disease labels to drive contrastive learning, enabling scalable training while preserving diagnostic granularity. Finally, to bridge the gap between pre-training and zero-shot inference and fully realize the potential of our disease-centric design, we introduce a diagnosis-aware prompt strategy. Rather than relying on handcrafted templates, we construct prompts from real clinical phrases and reuse the learned disease query tokens to extract consistent, condition-specific semantics. By aggregating these semantics across a diverse report collection into global disease prototypes, our approach significantly enhances inference robustness and boosts zero-shot diagnostic reliability.

We systematically evaluate our method across multiple

benchmarks. On the CT-RATE (Hamamci et al., 2024a) test set, our model achieves an AUC of 84.4%, outperforming the current state of the art by 5.1%. This gain remains consistent on the external Rad-ChestCT (Draelos et al., 2021) test set (+5.4% AUC), underscoring strong generalization. To further assess the scalability of disease-level alignment, we employ Qwen3-Max (Yang et al., 2025) to enrich CT-RATE with fine-grained disease annotations, resulting in a challenging diagnostic task spanning 60 distinct conditions. Under this more demanding setting, our approach exhibits an even more pronounced advantage, surpassing the strongest baseline by 9.8% AUC. Moreover, in the radiology report generation downstream, our pre-trained model demonstrates remarkable adaptability, further attesting to the generality and clinical relevance of the learned representations. Our contributions are summarized below:

• We propose a CNN-ViT hybrid encoder that replaces ViT's patch embedding with a 3D CNN and integrates multi-scale fusion, balancing efficiency and performance on 3D CT while retaining compatibility with pre-trained ViT priors.

• We introduce disease-level contrastive learning via learnable query tokens that dynamically extract and align disease-specific semantics from full reports—enabling fine-grained alignment without explicit report decomposition.

• We design a diagnosis-aware prompt strategy that uses real clinical descriptions and aggregated prototypes to bridge the training-inference gap and improve zero-shot diagnostic reliability.

## 2. Related Work

### 2.1. Medical Vision–Language Pre-training

VLP leverages radiology reports as rich textual supervision to align images and text in a shared semantic space. Early work focused on 2D chest X-rays using global image–report contrastive learning (Zhang et al., 2022; Boecking et al., 2022; Tiu et al., 2022; Chen et al., 2022; Huang et al., 2023; Cheng et al., 2023). Recent efforts extend VLP to 3D CT (Hamamci et al., 2024a; Cao et al., 2024; Blankemeier et al., 2026; Lin et al., 2024), but still rely on coarse global alignment. This is problematic in 3D CT, where critical pathologies occupy only a small fraction of the volume, causing models to be dominated by non-diagnostic background. Recent studies have explored complementary strategies to address data and modeling limitations: leveraging LLMs to generate silver-standard labels for supervised pre-training (Li et al., 2025), pre-training directly on uncurated multi-scan clinical studies via hierarchical attention (Zhao et al., 2025), and integrating auxiliary objectives such as report generation and masked image modeling to improve representation quality (Wald et al., 2025). While these advances enhance scalability and robustness, they still rely on global

image–report alignment and do not resolve the fundamental issue of semantic conflation among distinct pathologies within the same anatomical region.

### 2.2. Fine-grained Alignment in Medical VLP

To overcome the limitations of global alignment, fine-grained approaches have been proposed. In 2D, methods like GLoRIA (Huang et al., 2021) and LoVT (Müller et al., 2022) use cross-attention to implicitly align regions and sentences. However, such implicit mechanisms struggle with the complexity of 3D CT. fVLM (Shui et al., 2025) proposes explicit organ-level alignment by decomposing CT volumes and reports per organ, significantly improving performance. ViSD-Boost (Cao et al., 2025) identifies a "semantic density gap" between low-signal 3D images and high-signal reports, and enhances visual semantics by distinguishing normal from abnormal states per organ through contrastive learning and anatomical normality modeling via VQ-VAE(Van Den Oord et al., 2017). However, both methods treat all abnormalities within an organ as a single "abnormal" class, conflating distinct diseases. Our work advances beyond organ-level alignment by introducing disease-level contrastive learning, enabling disentanglement of multiple coexisting pathologies within the same organ.

## 3. Methodology

### 3.1. Vision-Language Pretraining Framework

As shown in Fig. 1, we present an end-to-end VLP framework that jointly optimizes anatomical segmentation, organ-level semantic alignment, and fine-grained disease-level representation learning. We named the model obtained from the pretrained framework as CT-DiagVLM.

**Visual Encoding.** Given an input 3D CT volume $\mathbf{V} \in \mathbb{R}^{D \times H \times W \times C}$, our CNN–ViT hybrid visual encoder $\text{Enc}_{\text{vis}}$ (detailed in Sec. 3.2) produces two complementary feature streams: (1) a multi-scale CNN feature hierarchy $\mathbf{F}_{\text{CNN}} = \{\mathbf{F}_1, \mathbf{F}_2, \mathbf{F}_3, \mathbf{F}_4\}$, extracted from the four stages of a 3D ResNet-18 (He et al., 2016) backbone, where $\mathbf{F}_i \in \mathbb{R}^{D_i \times H_i \times W_i \times C_i}$; and (2) a global ViT feature map $\mathbf{F}_{\text{ViT}} \in \mathbb{R}^{N \times d}$ with $N = D_4 H_4 W_4$, obtained through a ViT module (Dosovitskiy et al., 2021) that operates on patch embeddings derived from the CNN backbone. Formally,

$$\mathbf{F}_{\text{CNN}}, \mathbf{F}_{\text{ViT}} = \text{Enc}_{\text{vis}}(\mathbf{V}). \quad (1)$$

A U-Net-style (Ronneberger et al., 2015) segmentation decoder $\text{Dec}_{\text{seg}}$ takes only $\mathbf{F}_{\text{CNN}}$ as input and predicts $K$ soft organ masks $\{\hat{\mathbf{M}}_k\}_{k=1}^K$ at the original CT resolution. Each mask $\hat{\mathbf{M}}_k \in [0,1]^{D \times H \times W}$ is downsampled via max pooling to align with the ViT token grid, yielding $\mathbf{m}_k \in [0,1]^N$. The organ-specific visual embedding $\mathbf{v}_k$ is then obtained

by introducing learnable organ visual queries $\mathbf{q}_k^{\text{vis}} \in \mathbb{R}^d$ and computing cross-attention between $\mathbf{q}_k^{\text{vis}}$ and the mask-filtered ViT tokens:

$$\mathbf{v}_k = \text{CrossAttn}\big(\mathbf{q}_k^{\text{vis}}, \, \mathbf{m}_k \odot \mathbf{F}_{\text{ViT}}, \, \mathbf{m}_k \odot \mathbf{F}_{\text{ViT}}\big) \in \mathbb{R}^d, \quad (2)$$

where $\odot$ denotes element-wise multiplication.

To ensure anatomically plausible segmentations, we supervise $\text{Dec}_{\text{seg}}$ with a Dice loss (Sudre et al., 2017) against pseudo organ labels $\{\mathbf{M}_k^{\text{gt}}\}_{k=1}^K$ generated by TotalSegmentator (Wasserthal et al., 2023):

$$\mathcal{L}_{\text{dice}} = \frac{1}{K} \sum_{k=1}^K \left( 1 - \frac{2 \sum_i \hat{\mathbf{M}}_k(i) \cdot \mathbf{M}_k^{\text{gt}}(i)}{\|\hat{\mathbf{M}}_k\|_1 + \|\mathbf{M}_k^{\text{gt}}\|_1 + \epsilon} \right). \quad (3)$$

**Textual Encoding.** Following fVLM (Shui et al., 2025), we parse the radiology report $\mathcal{R}$ using an LLM into organ-specific descriptions $\{\mathcal{R}_k\}_{k=1}^K$. Each snippet $\mathcal{R}_k$ is tokenized and encoded by a transformer-based text encoder, yielding a sequence of contextual embeddings:

$$\mathbf{T}_k = \text{Enc}_{\text{lang}}\big(\text{Tokenize}(\mathcal{R}_k)\big) \in \mathbb{R}^{(L_k+1) \times d}, \quad (4)$$

where $L_k$ denotes the number of word tokens in $\mathcal{R}_k$, and the first token (i.e., the [CLS] token) $\mathbf{t}_k = \mathbf{T}_k[0] \in \mathbb{R}^d$ serves as the aggregate textual representation.

**Learning Objectives.** We adopt the same organ-level alignment strategy as fVLM (Shui et al., 2025), employing a batch-wise InfoNCE loss to align each organ's visual embedding $\mathbf{v}_k$ with its corresponding textual description $\mathbf{t}_k$:

$$\mathcal{L}_{\text{anat}} = \frac{1}{B * K} \sum_{b=1}^B \sum_{k=1}^K - \log \frac{\exp(\mathbf{v}_{b,k}^\top \mathbf{t}_{b,k}/\tau)}{\sum_{b'=1}^B \exp(\mathbf{v}_{b,k}^\top \mathbf{t}_{b',k}/\tau)}, \quad (5)$$

where both visual and textual embeddings are L2-normalized before computing similarity, $\tau$ is a temperature hyperparameter and $B$ denotes the batch size. More importantly, we introduce a disease-level contrastive learning mechanism that dynamically extracts disease-specific textual embeddings $\mathbf{s} \in \mathbb{R}^d$ from the organ descriptions $\{\mathcal{R}_k\}$, enabling fine-grained alignment between organ-level visual features and disease semantics. This yields an additional contrastive loss $\mathcal{L}_{\text{disease}}$ (formulated in Sec. 3.3).

The total training objective combines all components:

$$\mathcal{L}_{\text{total}} = \mathcal{L}_{\text{dice}} + \lambda_1 \mathcal{L}_{\text{anat}} + \lambda_2 \mathcal{L}_{\text{disease}}, \quad (6)$$

where $\lambda_1$ and $\lambda_2$ balances the contribution of organ-level and disease-level signals, and are empirically set to 0.5.

### 3.2. CNN–ViT Hybrid Encoder

Our hybrid encoder synergistically combines the local inductive bias of 3D CNNs with the global modeling capacity

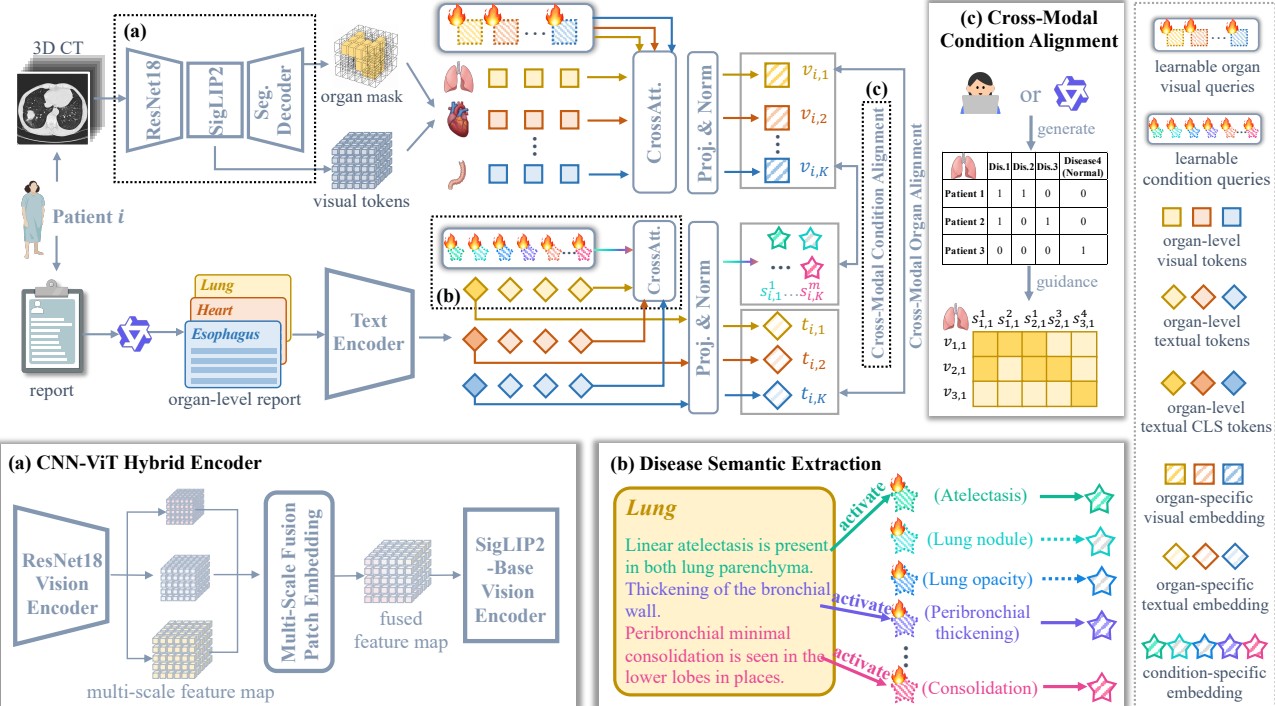

*Figure 1.* The proposed disease-centric vision–language pretraining framework. (a) CNN–ViT hybrid encoder replaces patch embedding with 3D ResNet-18 and MSF-PE to capture multi-scale anatomy while preserving ViT compatibility. (b) Learnable disease query tokens extract condition-specific semantics from full reports via cross-attention. (c) Organ-level visual features are aligned with disease-specific textual embeddings through fine-grained contrastive learning, disentangling coexisting pathologies within the same anatomical region.

of Vision Transformers (Dosovitskiy et al., 2021). Specifically, we retain a 3D ResNet-18 backbone (He et al., 2016) to extract hierarchical features $\mathbf{F}_{\text{CNN}} = \{\mathbf{F}_1, \mathbf{F}_2, \mathbf{F}_3, \mathbf{F}_4\}$, and replace ViT's native patch embedding module with a Multi-Scale Fusion Patch Embedding (MSF-PE) that fuses $\mathbf{F}_{\text{CNN}}$ into rich, multi-scale token initializations—enabling effective integration with pre-trained ViT weights.

Concretely, each CNN stage $\mathbf{F}_i$ is first projected to a common channel dimension $d$ via lightweight 3D convolutions:

$$\tilde{\mathbf{F}}_i = \text{Conv3D}_i(\mathbf{F}_i), \quad i = 1, 2, 3, 4. \tag{7}$$

Since $\mathbf{F}_4$ has the coarsest spatial resolution, we downsample $\tilde{\mathbf{F}}_1$, $\tilde{\mathbf{F}}_2$, and $\tilde{\mathbf{F}}_3$ using trilinear interpolation to align them with $\tilde{\mathbf{F}}_4$. The four aligned features are then concatenated along the channel dimension and fused through a final 3D convolution:

$$\mathbf{F}_{\text{fused}} = \text{Conv3D}_{\text{fuse}}([\tilde{\mathbf{F}}_1^{\downarrow}; \tilde{\mathbf{F}}_2^{\downarrow}; \tilde{\mathbf{F}}_3^{\downarrow}; \tilde{\mathbf{F}}_4]). \tag{8}$$

This fused volume $F_{\text{fused}} \in \mathbb{R}^{D_4 \times H_4 \times W_4 \times d}$ is flattened into a sequence of tokens, followed by LayerNorm and learnable 3D positional encoding to yield the final patch embeddings:

$$\mathbf{E}_{\text{patch}} = \text{LayerNorm}(\text{Flatten}(\mathbf{F}_{\text{fused}}) + \mathbf{P}_{\text{3D}}). \tag{9}$$

These embeddings are fed into a ViT encoder initialized with pre-trained SigLIP-2 weights (Tschannen et al., 2025),

producing the output feature map $\mathbf{F}_{\text{ViT}}$. This hybrid design effectively injects multi-scale local semantics from the CNN backbone into the transformer input, avoiding the computational burden of applying self-attention directly to high-resolution 3D volumes.

### 3.3. Disease-Level Contrastive Learning

While organ-level alignment provides coarse semantic grounding, it conflates distinct pathologies within the same anatomical structure. To enable fine-grained disease discrimination, we propose a disease-level contrastive learning mechanism that aligns organ-specific visual features with dynamically extracted condition-specific textual semantics across the entire batch.

**Condition Queries.** For each organ $k$, we define a set of learnable condition query vectors $\{q_k^{(c)}\}_{c \in \mathcal{C}_k}$, where $\mathcal{C}_k = \mathcal{M}_k \cup \{h\}$ denotes the set of all possible conditions for organ $k$: $\mathcal{M}_k$ is the set of diseases associated with organ $k$, and $h$ represents the healthy state. Consequently, the global condition set $\mathcal{C} = \{\mathcal{C}_1, \cdots, \mathcal{C}_K\}$ corresponds to the complete collection of diseases plus $K$ organ-specific healthy states. These queries are shared across all patients and optimized end-to-end.

**Dynamic Condition Semantic Extraction.** Given the full

textual embedding sequence $T_{b,k} \in \mathbb{R}^{(L_{b,k}+1) \times d}$ for organ $k$ of patient $b$, we extract condition-specific textual representations via cross-attention:

$$s_{b,k}^{(c)} = \text{CrossAttn}\big(q_k^{(c)}, T_{b,k}, T_{b,k}\big), \quad \forall c \in \mathcal{C}_k. \quad (10)$$

This allows the model to attend to relevant phrases describing either a specific disease or the normal state without requiring explicit report decomposition.

**Cross-Modal Condition Alignment.** Let $y_{b,k}^{(c)} \in \{0,1\}$ be a binary indicator (from ground-truth or LLM-generated pseudo-labels) denoting whether condition $c$ is present in organ $k$ of patient $b$. For each condition $c \in \mathcal{C}_k$, we construct a global set of positive textual embeddings across the batch:

$$\mathcal{S}_k^{(c)} = \left\{ s_{b',k}^{(c)} \,\middle|\, y_{b',k}^{(c)} = 1, \ b' \in [1, B] \right\}. \quad (11)$$

For any patient $b$ and organ $k$ diagnosed with condition $c$ (i.e., $y_{b,k}^{(c)} = 1$), we pull its organ-level visual embedding $v_{b,k}$ toward all textual embeddings in $\mathcal{S}^{(c)}$, while pushing it away from embeddings of other conditions. The disease-level contrastive loss is defined as:

$$\mathcal{L}_{\text{disease}} = \frac{1}{B * K} \sum_{b=1}^{B} \sum_{k=1}^{K} \sum_{c \in \mathcal{C}_k} \mathbb{I}[y_{b,k}^{(c)} = 1] \cdot \mathcal{L}_{b,k}^{(c)}, \quad (12)$$

where the per-instance loss $\mathcal{L}_{b,k}^{(c)}$ is given by

$$\mathcal{L}_{b,k}^{(c)} = -\log \frac{\sum_{s \in \mathcal{S}_k^{(c)}} \exp\left(v_{b,k}^{\top} s / \tau\right)}{\sum_{c' \in \mathcal{C}_k} \sum_{s' \in \mathcal{S}_k^{(c')}} \exp\left(v_{b,k}^{\top} s' / \tau\right)}, \quad (13)$$

with $\tau$ denoting the temperature and $\mathbb{I}[\cdot]$ the indicator function. This formulation enables consistent cross-patient alignment of identical diseases while explicitly separating different pathologies states—realizing true disease-centric semantic learning directly from unstructured clinical reports.

### 3.4. Diagnosis-Aware Prompt Strategy

In conventional zero-shot diagnosis, models typically rely on hand-crafted binary prompts such as "CT shows disease $m$" or "CT does not show disease $m$", which oversimplify clinical language and introduce a significant mismatch between pretraining and inference. To bridge this gap and fully realize the potential of our disease-centric design, we propose a diagnosis-aware prompt strategy that leverages real radiology descriptions as natural-language prompts.

Specifically, for each organ $k$ and condition $c \in \mathcal{C}_k = \mathcal{M}_k \cup \{h\}$, we construct a textual prototype by retrieving relevant reports from the training set. Retrieval is guided by ground-truth labels or LLM-generated pseudo-labels: reports indicating the presence of condition $c$ in

organ $k$ form the set $\mathcal{R}_k^{(c)}$. For each report $R \in \mathcal{R}_k^{(c)}$, let $T_R \in \mathbb{R}^{(L_R+1) \times d}$ denote its full textual embedding sequence, we then extract the condition-specific embedding using the same query-guided mechanism as in training:

$$s_R^{(c)} = \text{CrossAttn}\big(q_k^{(c)}, T_R, T_R\big), \quad \forall R \in \mathcal{R}_k^{(c)}. \quad (14)$$

This ensures semantic consistency between pretraining and inference.

To enhance robustness and reduce sensitivity to phrasing variations, we retrieve $M \geq 500$ reports per condition–organ pair. When fewer than $M$ positive samples exist (i.e., $|\mathcal{R}_k^{(c)}| < M$), we utilize all available reports. The final prototype is computed via global averaging:

$$p_k^{(c)} = \frac{1}{|\mathcal{R}_k^{(c)}|} \sum_{R \in \mathcal{R}_k^{(c)}} s_R^{(c)}. \quad (15)$$

This aggregation smooths out idiosyncratic details and captures the canonical semantic representation of each condition, while also mitigating label noise and enabling robust prototype learning even with LLM-generated pseudo-labels, as validated in Sec. 4.3.

During inference, diagnostic scores are obtained by computing cosine similarities between organ-level visual embeddings $v_k$ and the corresponding prototypes $\{\mathbf{p}_k^{(c)}\}_{c \in \mathcal{C}_k}$. By replacing artificial templates with real clinical language and leveraging query-guided semantic extraction, our approach achieves both semantic fidelity and operational robustness in zero-shot diagnosis.

## 4. Experiments

### 4.1. Experimental Setup

**Datasets.** We evaluate our method on two public chest CT benchmark datasets: CT-RATE (Hamamci et al., 2024a) and Rad-ChestCT (Draelos et al., 2021). CT-RATE contains 50,188 chest CT scans from 21,304 patients, each paired with a radiology report. Following the standard protocol, we use 20,000 patients for training and 1,304 for internal testing. Rad-ChestCT, comprising 3,630 CT volumes with radiology reports, serves as an external test set. Following prior works (Shui et al., 2025; Cao et al., 2025), we report performance on 16 diseases, excluding "Medical Material" and "Lymphadenopathy", to ensure a fair comparison. To further assess scalability and robustness under label noise, we use Qwen3-Max(Yang et al., 2025) to generate pseudo-labels for 60 diseases on CT-RATE.

**Baseline Methods.** Our primary baseline is fVLM (Shui et al., 2025), the current state-of-the-art organ-level vision–language pretraining (VLP) model for 3D CT. We find that both fVLM and our framework benefit from two minor

*Table 1.* Zero-shot performance comparison on the CT-RATE and Rad-ChestCT datasets.

| Method | Internal test (CT-RATE) | | | | External test (Rad-ChestCT) | | | |
|---|---|---|---|---|---|---|---|---|
| | AUC | ACC | F1 | Prec | AUC | ACC | F1 | Prec |
| Random (Hamamci et al., 2024a) | 50.5 | 50.2 | 57.0 | 18.0 | 49.6 | 50.0 | 55.5 | 26.5 |
| Supervised (Hamamci et al., 2024a) | 60.3 | 58.1 | 63.2 | 24.0 | 54.1 | 53.9 | 58.7 | 28.7 |
| CT-CLIP (Hamamci et al., 2024a) | 73.3 | 66.9 | 70.8 | 32.6 | 63.2 | 59.9 | 64.7 | 34.1 |
| CT-VocabFine (Hamamci et al., 2024a) | 76.0 | 70.4 | 73.8 | 35.6 | 65.7 | 62.1 | 66.8 | 35.6 |
| CT-LiPro (Hamamci et al., 2024a) | 76.1 | 69.1 | 72.6 | 34.3 | 64.7 | 60.6 | 65.0 | 35.1 |
| BIUD (Cao et al., 2024) | 71.3 | 68.1 | 71.6 | 33.8 | 62.9 | 60.6 | 65.2 | 33.7 |
| Merlin (Blankemeier et al., 2026) | 72.8 | 67.2 | 70.9 | 33.7 | 64.4 | 61.9 | 66.3 | 34.8 |
| HLIP (Zhao et al., 2025) | 77.7 | 71.4 | 74.7 | 37.9 | 72.3 | 68.4 | 72.1 | 40.4 |
| COLIPRI (Wald et al., 2025) | 77.8 | 70.2 | 75.2 | - | 67.0 | 60.9 | 65.7 | - |
| fVLM (Shui et al., 2025) | 77.8 | 71.8 | 75.1 | 37.9 | 68.0 | 64.7 | 68.8 | 37.4 |
| ViSD-Boost (Cao et al., 2025) | 79.0 | 73.1 | 75.9 | 38.7 | 69.4 | 65.2 | 69.3 | 34.2 |
| Base Model | 79.3 | 74.6 | 77.3 | 41.9 | 70.0 | 64.5 | 68.7 | 34.5 |
| CT-DiagVLM | **84.4** | **78.1** | **80.3** | **45.2** | **75.4** | **69.8** | **73.4** | **39.2** |

yet impactful adaptations: (1) adopting a 3D ResNet-18 backbone (He et al., 2016) to better preserve local anatomical details in volumetric data; and (2) replacing offline organ masks from TotalSegmentator (Wasserthal et al., 2023) with online masks predicted by an auxiliary U-Net decoder (Sec. 3.1), enabling end-to-end optimization. To ensure a fair comparison, we integrate these into fVLM to form a stronger *base model*, which already surpasses the original and achieves state-of-the-art performance. Crucially, our experiments show that our proposed framework delivers further substantial gains on top of this strong baseline.

**Evaluation Metrics.** We adopt standard metrics for zero-shot diagnostic performance: Area Under the ROC Curve (AUC), Accuracy (ACC), Sensitivity (Sens), Specificity (Spec), F1-score, and Precision (Prec). To ensure fair comparison, we follow the evaluation protocol of prior works (Shui et al., 2025; Cao et al., 2025), selecting the decision threshold for each disease by minimizing the Euclidean distance to the ideal point (0,1) on the ROC curve. For the report-generation task, we evaluate using both diagnostic metrics (e.g., disease detection accuracy) and natural language generation metrics (e.g., BLEU, METEOR).

### 4.2. Zero-shot Diagnosis on Standard Benchmarks

We evaluate the zero-shot diagnostic performance of our model on both the internal CT-RATE test set and the external Rad-ChestCT dataset, comparing against recent state-of-the-art methods including fVLM (Shui et al., 2025) and ViSD-Boost (Cao et al., 2025). As shown in Tab. 1, our method achieves an AUC of 84.4% on CT-RATE, outperforming fVLM (77.8%) by a substantial margin of 6.6% and surpassing ViSD-Boost (79.0%) by 5.4%. This trend is

*Table 2.* Zero-shot performance comparison on the CT-RATE dataset using pseudo-labels for 60 diseases.

| Metric | Base Model | CT-DiagVLM |
|---|---|---|
| AUC | 75.8 | **85.6** |
| ACC | 73.8 | **85.3** |
| F1 | 78.6 | **87.0** |
| Prec | 27.8 | **37.3** |
| Spec | 74.0 | **88.8** |
| Sens | **72.0** | 65.5 |

consistently observed across all evaluation metrics—ACC, F1, and Precision—demonstrating the robustness of our disease-centric alignment strategy.

More importantly, our model exhibits strong generalization to unseen data distributions. On the external Rad-ChestCT test set, our approach attains an AUC of 75.4%, improving upon fVLM (68.0%) by 7.4% and ViSD-Boost (69.4%) by 6.0%. These results validate that our hybrid visual encoder, disease-level contrastive learning, and diagnosis-aware prompt strategy collectively enable more precise and generalizable vision–language alignment than existing organ-level or semantic-density-based approaches.

To further mitigate the impact of label exposure during pre-training and ensure a more rigorous "zero-shot" assessment, we conduct a strict hold-out experiment in which a disjoint subset of pathologies is entirely withheld from training and reserved exclusively for testing. As detailed in Appendix A.4, our model maintains robust diagnostic performance on these truly unseen diseases, demonstrating strong generalization capabilities that extend well beyond

the pre-trained label space.

### 4.3. Zero-shot Diagnosis When Scaling to 60 Diseases

To evaluate the scalability and robustness of our approach under label noise, we leverage Qwen3-Max (Yang et al., 2025) to generate pseudo-labels for 60 diseases on CT-RATE (details in Appendix A.2). As shown in Tab. 2, our full model achieves an AUC of 85.6%, substantially outperforming the base model (75.8%) by 9.8%. This significant gain demonstrates that our method generalizes effectively to a much broader diagnostic spectrum.

Notably, while AUC and specificity improve substantially, sensitivity declines from 72.0% to 65.5%. This shift reflects the inherent calibration of our diagnosis-aware prompting strategy: by aggregating semantic cues from diverse clinical reports to construct disease prototypes, the model learns tighter decision boundaries that prioritize findings with strong alignment to canonical disease manifestations, thereby suppressing ambiguous positives. This conservative behavior inherently enhances robustness to label noise, as evidenced by stable AUC performance even for conditions with the noisiest pseudo-labels (Appendix A.6). Importantly, the sensitivity reduction represents a deliberate and controllable trade-off rather than a model limitation; it can be readily mitigated by incorporating handcrafted templates, which restore sensitivity to 78.6% without retraining (Appendix A.6). Furthermore, as detailed in Appendix A.7, our method achieves robust performance on rare conditions with limited training samples in CT-RATE (e.g., "honeycombing" "andpneumothorax"). This indicates that the condition query–based design effectively preserves discriminative signals for low-prevalence pathologies throughout training.

### 4.4. Radiology Report Generation

To validate the generality and clinical utility of the learned representations, We adapt CT-DiagVLM to the downstream task of radiology report generation by appending a BERT-base (Devlin et al., 2019) text decoder. Given that traditional NLP metrics (e.g., BLEU, METEOR) poorly reflect clinical value, we follow CT-CHAT (Hamamci et al., 2024a) by using an auxiliary labeler to extract structured abnormality labels from generated reports and evaluate performance using clinical efficacy metrics—Precision, Recall, and F1-score. As shown in Tab. 3, our model significantly outperforms existing state-of-the-art methods on both the internal and external test sets, achieving an F1-score of 45.5% on CT-RATE, and an F1-score of 37.3% on Rad-ChestCT. These results further corroborate the strong generalization capability of the semantic space learned by our framework.

### 4.5. Ablation Study

**Contributions of Each Components.** We conduct an ablation study on CT-RATE to quantify the contribution of each proposed component (Tab. 4). Starting from our base model (79.3% AUC), replacing the plain CNN encoder with our CNN–ViT hybrid architecture yields a consistent gain of +1.0% AUC (to 80.3%). Notably, this improvement is orthogonal and consistently present: even when all other components (prompt strategy and disease-level loss) are enabled, swapping the CNN encoder for the hybrid variant still provides an additional +1.2% AUC (from 83.2% to 84.4%). Building upon this hybrid encoder, the introduction of our diagnosis-aware prompt strategy further improves performance by +2.1% AUC (from 80.3% to 82.4%). Finally, the integration of disease-level contrastive learning brings an additional +2.0% AUC, leading to the full model's 84.4% AUC (+5.1% over the base). These results collectively validate that each component contributes uniquely and synergistically to the final performance.

**Hybrid Encoder Design.** We further investigate architectural and initialization strategies for our CNN–ViT hybrid encoder, with results summarized in Tab. 4. First, we explore advanced representation enhancement techniques built upon the hybrid backbone, including (1) DeepStack (Meng et al., 2024): stacking multiple CNN blocks in different ViT layers to deepen feature interaction; and (2) Organ-level MoE (Jacobs et al., 1991; Shazeer et al., 2017): routing organ-specific visual tokens through a mixture-of-experts layer to capture anatomical heterogeneity. Surprisingly, neither strategy yields consistent gains—DeepStack even degrades performance slightly (84.0% vs. 84.4%). This suggests that our current design is already sufficient to extract rich and discriminative representations.

Second, we validate the importance of leveraging large-scale pre-trained ViT knowledge. Initializing the ViT component from scratch results in a significant drop in AUC (76.5%), confirming that pre-training priors are crucial. Both general-purpose (SigLIP2 Base/So-400M (Tschannen et al., 2025)) and medical-domain (MedSigLIP So-400M (Sellergren et al., 2025)) pre-trained checkpoints substantially outperform the variant equipped with 3D ResNet-18 backbone (83.2% AUC), demonstrating the transferability of vision-language priors —even when trained on non-medical data. The SigLIP2 Base initialization achieves the best performance (84.4% AUC) among all variants. Thus, we adopt this configuration as our default hybrid encoder.

**Prompt Strategy Design.** We further investigate different strategies for constructing textual prompts during zero-shot inference. As shown in Tab. 4 (row 3), we first explore a handcrafted prompt refinement approach, where for each disease we manually select a representative clinical phrase based on expert knowledge. While this yields an improve-

*Table 3.* Report generation performance. We evaluate using both diagnostic metrics (e.g., disease detection accuracy) and natural language generation metrics (e.g., BLEU, METEOR). BLEU and METEOR scores for the baseline method are taken from (Hamamci et al., 2025).

| Method | Internal test (CT-RATE) | | | | | External test (Rad-ChestCT) | | |
|---|---|---|---|---|---|---|---|---|
| | BLEU-2 | METEOR | Prec | Recall | F1 | Prec | Recall | F1 |
| RadFM (Wu et al., 2025) | 29.2 | 19.7 | 17.3 | 4.1 | 6.1 | 25.9 | 5.1 | 7.9 |
| CT2Rep (Hamamci et al., 2024b) | 16.3 | 14.8 | 45.8 | 14.3 | 18.0 | 31.2 | 14.8 | 13.0 |
| CT-CHAT (Hamamci et al., 2024a) | 28.4 | **21.5** | 41.1 | 17.6 | 20.2 | 39.3 | 17.0 | 17.0 |
| CT-DiagVLM | **33.4** | 20.9 | **55.6** | **39.9** | **45.5** | **42.7** | **38.5** | **37.3** |

*Table 4.* Ablation study of key components in CT-DiagVLM on the CT-RATE dataset. ▲ denotes the intermediate version designed in our ablation study, whose details will be discussed in Sec. 4.5.

| Method | Hybrid Encoder | Disease-level Contrastive Loss | Diagnosis-Aware Prompt | AUC |
|---|---|---|---|---|
| Base Model | ✗ | ✗ | ✗ | 79.3 |
| CT-DiagVLM | ✓ | ✗ | ✗ | 80.3 |
| | ✓ | ✗ | ▲ | 81.0 |
| | ✓ | ✗ | ✓ | 82.4 |
| | ✓ | ▲ | ✓ | 83.2 |
| | ✗ | ✓ | ✓ | 83.2 |
| | ✓ | ✓ | ✓ | 84.4 |

*Table 5.* Ablation study on different design choices of the hybrid encoder.

| Method | AUC |
|---|---|
| CT-DiagVLM | 84.4 |
| *Representation Optimization Strategies:* | |
| + DeepStack (Meng et al., 2024) | 84.0 |
| + Organ-level MoE (Shazeer et al., 2017) | 84.4 |
| *The Size and Initialization Strategy of ViT:* | |
| Scratch | 76.5 |
| SigLIP2 Base (Tschannen et al., 2025) | 84.4 |
| SigLIP2 So-400M (Tschannen et al., 2025) | 84.2 |
| MedSigLIP So-400M (Sellergren et al., 2025) | 83.8 |

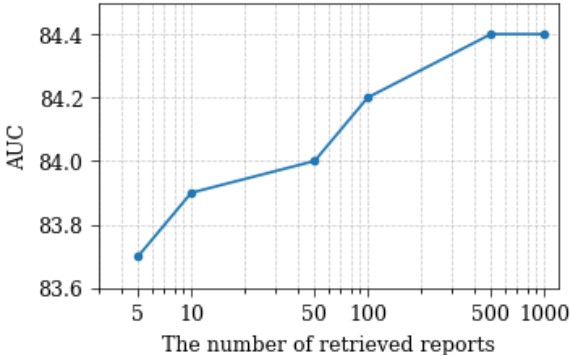

*Figure 2.* Ablation on the number of retrieved reports $M$.

ment (+0.7% AUC), it is outperformed by our automated diagnosis-aware prompt strategy (+2.1% AUC, row 4). This highlights the advantage of leveraging diverse, real-world report language over fixed templates, as the latter cannot capture the natural variability in radiological descriptions.

To study the impact of report diversity on prototype robustness, we vary the number of retrieved reports $M$ used to construct each disease prototype. As illustrated in Fig. 2, performance is suboptimal when $M$ is small (e.g., 5 or 10) due to high variance and label noise in limited samples. The AUC steadily improves as $M$ increases, plateauing at $M \geq 500$, where our model achieves its peak performance of 84.4%. This confirms that aggregating semantics from a

large and diverse set of clinical reports is crucial for building stable and reliable disease prototypes.

**Disease-level Contrastive Loss Design.** We further investigate the granularity of contrastive alignment in the disease-level learning objective. We explore a coarse disease-aware strategy as an intermediate design: for any two patients, if they share identical disease annotations within a given organ (e.g., both have "emphysema" and "lung nodule" in the lungs), we treat their organ-level image–text pairs as positive and pull them together. This approach goes beyond pure organ-level matching by considering coarse disease co-occurrence, yet stops short of modeling individual pathologies explicitly. As shown in Tab. 4 (row 5), this intermediate strategy achieves 83.2% AUC—improving over

Arterial wall calcification

Pleural effusion

Hiatal hernia

Cardiomegaly

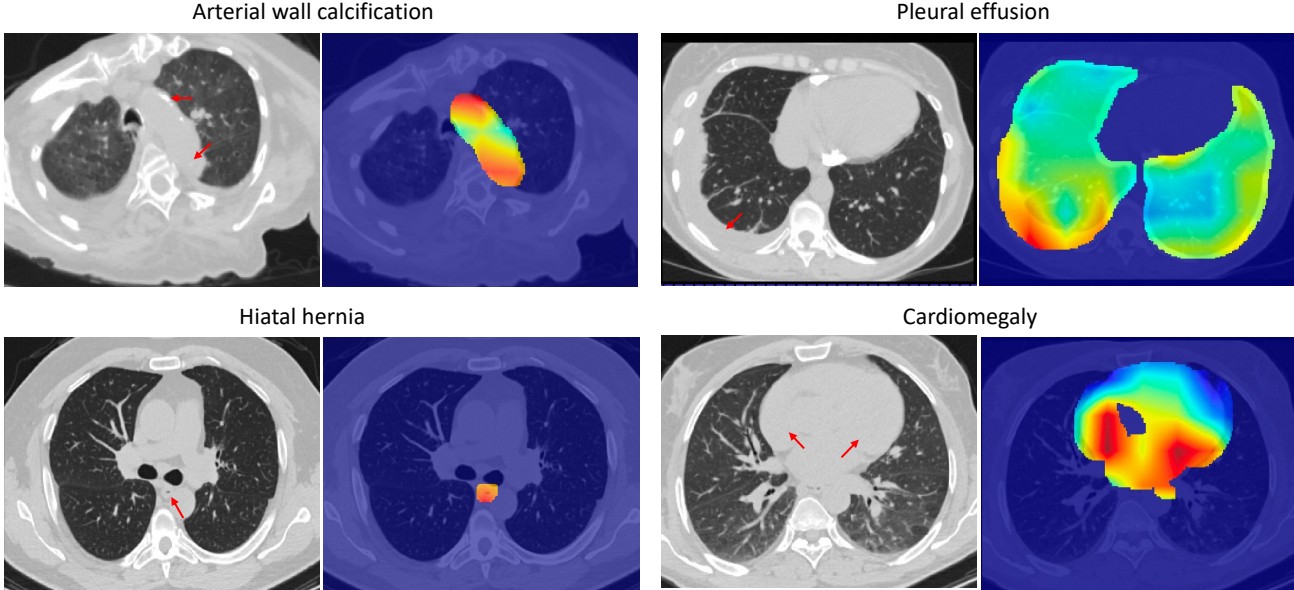

*Figure 3.* Visualization of the activation maps generated by our model.

the variant with only organ-level alignment (row 4, 82.4%) by +0.8%. However, it still falls short of our full disease-level contrastive learning (row 7, 84.4%), which explicitly aligns visual features with individual disease semantics. The additional gain underscores that fine-grained, per-disease alignment is essential for disentangling coexisting pathologies and achieving precise cross-modal semantic grounding.

### 4.6. Visualization

We visualize the activation maps generated by our model to qualitatively assess the localization capability of our model. As shown in Figure 3, our model successfully highlights pathological regions corresponding to specific diseases without any disease-level bounding box or segmentation mask supervision during training. This emergent localization ability stems from the fine-grained cross-modal alignment between visual tokens and disease-specific textual semantics, demonstrating that disease-level contrastive learning not only improves diagnostic accuracy but also enhances interpretability through implicit lesion localization.

## 5. Conclusion

In this work, we present a disease-centric vision–language pretraining framework tailored for 3D CT imaging. By co-designing an efficient CNN–ViT hybrid encoder, fine-grained disease-level contrastive learning, and a diagnosis-aware prompt strategy, our approach overcomes two key limitations of existing medical VLP models: the inefficiency of visual backbones on 3D data and the coarse semantic alignment that conflates distinct pathologies within

the same anatomical region. The obtained CT-DiagVLM model achieves state-of-the-art performance on both the CT-RATE and Rad-ChestCT benchmarks, with +5.1% and +5.4% AUC gains over the strongest prior method. Under a challenging 60-disease setting with LLM-generated pseudo-labels, it shows even larger improvements (+9.8% AUC), demonstrating robustness to label noise and scalability across broader diagnostic spectra. Its strong transferability is further confirmed by superior radiology report generation, highlighting the generality and clinical relevance of the learned representations. We believe our work provides a promising direction toward clinically viable, general-purpose medical AI systems that diagnose accurately and reason with the nuance of real clinical language.

**Limitation.** CT-DiagVLM relies on external tools for report parsing and pseudo mask/label generation. While most pseudo-annotations are reliable, residual noise may still affect representation learning. Furthermore, we frame this work as "pretraining" since our goal is to learn task-agnostic, disease-level representations. Although we validate the learned features through multiple evaluations, broader assessment remains constrained by the current lack of standardized downstream benchmarks in medical AI. We plan to expand evaluation as community benchmarks mature.

## Acknowledgements

This work was supported in part by the National Natural Science Foundation of China under Grants 62431017, 62320106003, 62371288, in part by the Fundamental and Interdisciplinary Disciplines Breakthrough Plan of the Min-

istry of Education of China (JYB2025XDXM611), and in part by AI for Science Program, Shanghai Municipal Commission of Economy and Informatization under Grant 2025-GZL-RGZN-BTBX-02022. Jianpeng Zhang was supported by the Zhejiang Province Postdoctoral Research Excellence Funding Program (ZJ2024032).

## Impact Statement

This paper presents work whose goal is to advance the field of Machine Learning. There are many potential societal consequences of our work, none which we feel must be specifically highlighted here.

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

# A. Appendix

## A.1. Implementation Details.

All CT volumes are resampled to an isotropic resolution of $1\,\text{mm} \times 1\,\text{mm} \times 5\,\text{mm}$. Hounsfield Unit (HU) values are clipped to the range $[-1150, 350]$, a window optimized for chest tissue contrast, and then normalized to $[0, 1]$. During training, we apply random cropping with a fixed patch size of $112 \times 256 \times 384$ along the axial, coronal, and sagittal axes, respectively. Only anatomical structures fully contained within the cropped volume are used for alignment to preserve semantic integrity. Our vision encoder employs the proposed CNN–ViT hybrid architecture (Sec. 3.2), the text encoder utilizes CXR-BERT (Boecking et al., 2022), and the learnable query tokens are initialized via truncated normal distribution. The model is trained using the Adam optimizer with a cosine-decay learning rate schedule, peaking at $1 \times 10^{-4}$. We train for 30 epochs on 12 H20 GPUs with a total batch size of 72.

## A.2. Details for Disease Label Expansion

As mentioned in Sec. 4.1, to evaluate the scalability and robustness of our disease-centric vision–language pretraining framework under fine-grained diagnostic settings, we expanded the original CT-RATE label space from 16 coarse disease categories to a comprehensive set of 60 distinct pathological conditions. This expansion process involved the following three steps.

• **Disease Taxonomy Construction.** We systematically reviewed the category space defined in the authoritative literature (Draelos et al., 2021) and constructed a fine-grained disease taxonomy comprising 60 distinct conditions, specifically adapted to the chest CT diagnostic scenario.

• **Automated Disease Label Extraction.** To enable large-scale data annotation, we employed Qwen3-Max (Yang et al., 2025) and used the following prompt to automatically extract the aforementioned 60 disease labels from unstructured radiology reports, generating pseudo-labels for model training.

---

**Prompt for Disease Label Extraction**

**Radiology Report:** `<report>`

Please determine whether this CT report indicates that the patient has or is suspected of having `<disease>` in the `<organ>` region. Respond ONLY with "Yes" or "No".

**Instructions:**
- If the report uses uncertain terms such as "considered", "suspicious for", "suggestive of", "possible", or "not excluded" regarding this disease, answer "Yes".
- If only an anatomical abnormality is mentioned without linking to this specific disease, answer "No".
- Your response MUST start with either "Yes" or "No". Do not add any explanation.

---

• **Label Quality Assessment and Refinement.** We trained a BERT-based (Devlin et al., 2019) report-to-label classifier using the extracted labels and evaluated the annotation accuracy on a test set. Subsequently, we leveraged the classifier's predictions to identify and correct potentially erroneous labels, thereby enhancing the overall reliability of the final label set.

The proposed labeling pipeline deliberately omits manual curation for the training data. This design choice aims to evaluate the model under a realistic and scalable scenario, where labels are generally accurate but inevitably contain non-negligible noise. Such pseudo-labels are readily obtainable in the era of large language models (LLMs) and closely mirror practical deployment conditions, where perfectly curated annotations are rarely available. To ensure evaluation reliability, however, we enlisted resident physicians to manually verify the test set labels. By benchmarking the LLM-generated labels against this physician-verified reference using a BERT classifier, empirical results indicate that the pseudo-labels produced by Qwen3-Max in Step 2 achieve 97.2% accuracy and a 77.9% F1-score. Notably, for diseases with more than 500 training samples, the F1-score improves to 83.0%, which adequately meets the requirements of our experimental design.

*Table 6.* AUC performance on held-out diseases under different training label configurations.

| Method | Training Labels | AUC on Held-out 6 Diseases |
|---|---|---|
| Base Model | Full 16 Diseases | 81.4 |
| CT-DiagVLM | 10 Diseases Only | **83.1** |

*Table 7.* Diagnostic performance of CT-DiagVLM under zero-shot and fine-tuned settings.

| Method | AUC | ACC | F1 |
|---|---|---|---|
| CT-DiagVLM (zero-shot) | 84.4 | 78.1 | 80.3 |
| + Fine-tuning | **86.0** | **79.4** | **81.4** |

## A.3. Quality Assessment of Radiology Report Generation

As mentioned in Sec. 4.4, to quantitatively evaluate the clinical fidelity of generated radiology reports, we adopt diagnostic metrics—namely Precision, Recall, and F1-score—based on disease-level correctness. Specifically, we employ a 16-class report classifier, trained following the same procedure as described in Step 3 of the label expansion pipeline (Appendix A.2), to extract predicted disease labels from the model-generated reports. The diagnostic metrics are then computed by comparing these extracted labels against the ground-truth disease labels provided by the dataset, offering an objective measure of how accurately the generated text reflects the true underlying pathological findings.

## A.4. Hold-out Experiment

We conducted a strict hold-out experiment to evaluate performance on unseen diseases and demonstrate the robust cross-disease generalization abilitiy of CT-DiagVLM. We pre-trained on only 10 out of 16 CT-RATE diseases, explicitly excluding 5 lung diseases (last 5 indices) and 1 heart disease (1st index). On the held-out 6 diseases, we evaluated performance without condition-specific query tokens, relying solely on organ-level feature similarity. As shown in Tab. 6, our model achieves 83.1% AUC on these unseen diseases, surpassing the Base Model by +1.7%. This confirms that our disease-level contrastive learning learns transferable fine-grained semantics even without explicit training labels.

## A.5. Continue Finetuning Results

To further evaluate the adaptability of our pre-trained representations, we perform supervised fine-tuning on the CT-RATE training set. As shown in Tab. 7, fine-tuning yields consistent performance gains, improving AUC, ACC, and F1 by 1.6%, 1.3%, and 1.1%, respectively. This result underscores the practical flexibility of our framework, which supports direct zero-shot deployment while retaining strong fine-tuning potential when task-specific annotations are available.

## A.6. The Robustness of CT-DiagVLM to Label Noise

To empirically validate the robustness of our approach to label noise, we analyze performance across diseases with varying pseudo-label quality. Specifically, we identify the top-10 diseases with the lowest and highest label noise (measured by Qwen3-Max pseudo-label F1-scores). As shown in Tab. 8, AUC remains stable across both groups despite reduced sensitivity for high-noise diseases. Complementarily, Tab. 9 demonstrates that removing diagnosis-aware prompting consistently degrades multiple metrics (AUC, ACC, F1, precision, specificity) under pseudo-label supervision. Together, these results confirm the discussion in Sec. 4.3 that by aggregating semantics from diverse clinical reports, our diagnosis-aware prompt strategy yields more precise and robust disease diagnosis under label noise. Notably, even in the absence of diagnosis-aware prompting, the combination of our CNN–ViT hybrid encoder and disease-level contrastive learning yields an 81.1% AUC and an improved 78.8% sensitivity. This markedly surpasses the base model's 75.8% AUC under the same pseudo-label conditions, demonstrating the intrinsic effectiveness of our disease-centric pretraining architecture in learning high-quality representations.

## A.7. Details about Zero-shot Performance

Tab. 10 and Tab. 11 provide a per-disease breakdown of the CT-RATE results reported in Tab. 1 and Tab. 2, respectively, for our CT-DiagVLM model. These detailed results illustrate the model's consistent performance across a wide spectrum of

*Table 8.* Average performance metrics for the top-10 diseases with the lowest and highest label noise.

| Group | AUC | ACC | Sens | Spec |
|---|---|---|---|---|
| Low F1-Score (Top 10 lowest) | 0.815 | 0.863 | 0.498 | 0.899 |
| High F1-Score (Top 10 highest) | 0.900 | 0.848 | 0.799 | 0.880 |

*Table 9.* Ablation study on diagnosis-aware prompting on the CT-RATE dataset using pseudo-labels for 60 diseases.

| Metric | CT-DiagVLM w/o Diagnosis-Aware Prompt | CT-DiagVLM |
|---|---|---|
| AUC | 81.1 | **85.6** |
| ACC | 75.5 | **85.3** |
| F1 | 80.4 | **87.0** |
| Prec | 29.4 | **37.3** |
| Spec | 75.5 | **88.8** |
| Sens | **78.6** | 65.5 |

*Table 10.* Disease-specific zero-shot performance of CT-DiagVLM on the CT-RATE dataset.

| Organ | Disease | AUC | ACC | Sens | Spec |
|---|---|---|---|---|---|
| lung | Pleural effusion | 0.965 | 0.945 | 0.932 | 0.946 |
| aorta | Arterial wall calcification | 0.940 | 0.884 | 0.914 | 0.871 |
| heart | Cardiomegaly | 0.937 | 0.861 | 0.891 | 0.858 |
| heart | Coronary artery wall calcification | 0.937 | 0.879 | 0.915 | 0.866 |
| heart | Pericardial effusion | 0.900 | 0.853 | 0.816 | 0.856 |
| lung | Consolidation | 0.892 | 0.802 | 0.871 | 0.785 |
| lung | Interlobular septal thickening | 0.878 | 0.787 | 0.813 | 0.784 |
| lung | Mosaic attenuation pattern | 0.860 | 0.793 | 0.784 | 0.794 |
| esophagus | Hiatal hernia | 0.843 | 0.773 | 0.735 | 0.780 |
| lung | Lung opacity | 0.833 | 0.762 | 0.719 | 0.790 |
| lung | Peribronchial thickening | 0.797 | 0.718 | 0.795 | 0.708 |
| lung | Emphysema | 0.786 | 0.702 | 0.737 | 0.694 |
| lung | Bronchiectasis | 0.784 | 0.729 | 0.736 | 0.728 |
| lung | Atelectasis | 0.772 | 0.712 | 0.711 | 0.712 |
| lung | Pulmonary fibrotic sequela | 0.719 | 0.662 | 0.673 | 0.641 |
| lung | Lung nodule | 0.668 | 0.635 | 0.607 | 0.659 |

pathologies under both the standard 16-disease setting and the more challenging 60-disease scenario with LLM-generated pseudo-labels. Tab. 12 further summarizes the 60 diseases covered by our taxonomy and lists the number of positive samples for each disease in the CT-RATE training set after applying the refinement procedure described in Step 3 of Appendix A.2. Notably, as evidenced by the joint analysis of Tab. 11 and Tab. 12, CT-DiagVLM maintains robust diagnostic performance on rare conditions despite their scarcity in the training set—for instance, "honeycombing" (186 positive cases) and "pneumothorax" (127 positive cases) achieve AUCs of 0.910 and 0.942, respectively. This resilience stems from our condition query–based design, which explicitly preserves discriminative signals from low-prevalence pathologies during training, preventing their sparse evidence from being overwhelmed by more common diseases. This capability is crucial for real-world clinical deployment, where accurate detection of low-prevalence but high-impact pathologies is essential.

*Table 11.* Disease-specific zero-shot performance of CT-DiagVLM on the CT-RATE dataset using pseudo-labels for 60 diseases.

| Organ | Disease | AUC | ACC | Sens | Spec |
|---|---|---|---|---|---|
| aorta | stent | 0.999 | 0.996 | 1.000 | 0.996 |
| esophagus | GI tube | 0.999 | 0.993 | 1.000 | 0.993 |
| heart | transplant | 0.998 | 0.994 | 1.000 | 0.994 |
| heart | heart valve replacement | 0.993 | 0.956 | 1.000 | 0.956 |
| heart | pacemaker or defibrillator | 0.974 | 0.973 | 0.886 | 0.974 |
| lung | pleural effusion | 0.966 | 0.944 | 0.930 | 0.945 |
| heart | congestion | 0.964 | 0.919 | 0.859 | 0.921 |
| heart | CABG | 0.956 | 0.908 | 0.877 | 0.909 |
| heart | coronary artery disease | 0.955 | 0.898 | 0.911 | 0.893 |
| heart | heart failure | 0.951 | 0.903 | 0.855 | 0.907 |
| aorta | atherosclerosis | 0.948 | 0.894 | 0.917 | 0.885 |
| aorta | plaque | 0.945 | 0.892 | 0.907 | 0.886 |
| aorta | calcification | 0.944 | 0.891 | 0.904 | 0.886 |
| lung | pneumothorax | 0.942 | 0.889 | 0.929 | 0.889 |
| heart | cardiomegaly | 0.937 | 0.879 | 0.845 | 0.883 |
| heart | sternotomy | 0.934 | 0.927 | 0.843 | 0.929 |
| lung | pulmonary edema | 0.918 | 0.935 | 0.686 | 0.944 |
| lung | honeycombing | 0.910 | 0.822 | 0.750 | 0.823 |
| lung | pneumonia | 0.897 | 0.834 | 0.660 | 0.945 |
| lung | cavitation | 0.896 | 0.863 | 0.667 | 0.865 |
| lung | infiltrate | 0.893 | 0.814 | 0.615 | 0.960 |
| lung | pneumonitis | 0.893 | 0.836 | 0.658 | 0.940 |
| heart | pericardial effusion | 0.892 | 0.821 | 0.850 | 0.819 |
| lung | infection | 0.892 | 0.824 | 0.677 | 0.929 |
| aorta | aneurysm | 0.891 | 0.822 | 0.812 | 0.823 |
| lung | airspace disease | 0.891 | 0.819 | 0.649 | 0.941 |
| lung | consolidation | 0.887 | 0.807 | 0.844 | 0.798 |
| lung | lung resection | 0.885 | 0.853 | 0.667 | 0.854 |
| lung | postsurgical | 0.864 | 0.884 | 0.538 | 0.887 |
| lung | opacity | 0.860 | 0.777 | 0.768 | 0.792 |
| lung | nodule>1cm | 0.852 | 0.878 | 0.619 | 0.895 |
| lung | groundglass | 0.846 | 0.808 | 0.658 | 0.890 |
| lung | tree in bud | 0.823 | 0.858 | 0.616 | 0.872 |
| lung | cancer | 0.821 | 0.863 | 0.475 | 0.901 |
| lung | septal thickening | 0.814 | 0.842 | 0.381 | 0.893 |
| esophagus | lesion | 0.811 | 0.869 | 0.750 | 0.870 |
| lung | aspiration | 0.801 | 0.901 | 0.250 | 0.916 |
| heart | pericardial thickening | 0.792 | 0.879 | 0.571 | 0.881 |
| lung | tuberculosis | 0.789 | 0.863 | 0.545 | 0.885 |
| lung | mucous plugging | 0.778 | 0.859 | 0.467 | 0.865 |
| lung | air trapping | 0.777 | 0.809 | 0.520 | 0.835 |
| lung | bronchiolectasis | 0.777 | 0.842 | 0.523 | 0.874 |
| lung | bronchial wall thickening | 0.776 | 0.831 | 0.404 | 0.891 |
| lung | mass | 0.775 | 0.853 | 0.416 | 0.900 |
| lung | emphysema | 0.771 | 0.725 | 0.631 | 0.749 |
| lung | pleural thickening | 0.767 | 0.852 | 0.432 | 0.874 |
| lung | atelectasis | 0.763 | 0.736 | 0.499 | 0.828 |
| lung | bronchiectasis | 0.754 | 0.814 | 0.463 | 0.863 |
| lung | bronchitis | 0.739 | 0.840 | 0.300 | 0.894 |
| lung | bronchiolitis | 0.734 | 0.820 | 0.211 | 0.912 |
| lung | interstitial lung disease | 0.718 | 0.750 | 0.270 | 0.890 |
| lung | reticulation | 0.712 | 0.846 | 0.238 | 0.899 |
| lung | fibrosis | 0.687 | 0.764 | 0.397 | 0.830 |
| lung | scattered nodules/nodes | 0.680 | 0.641 | 0.275 | 0.901 |
| lung | nodule | 0.668 | 0.626 | 0.608 | 0.643 |

*Table 12.* The 60 diseases encompassed by our taxonomy and the number of positive samples for each disease in the CT-RATE training set.

| Organ | Disease | Disease count |
|---|---|---|
| aorta | aneurysm | 1179 |
| aorta | atherosclerosis | 6462 |
| aorta | calcification | 6180 |
| aorta | plaque | 6331 |
| aorta | stent | 82 |
| esophagus | GI tube | 126 |
| esophagus | cancer | 51 |
| esophagus | debris | 18 |
| esophagus | distention | 55 |
| esophagus | lesion | 130 |
| esophagus | mass | 31 |
| heart | CABG | 721 |
| heart | cardiomegaly | 2602 |
| heart | congestion | 848 |
| heart | coronary artery disease | 6110 |
| heart | heart failure | 1887 |
| heart | heart valve replacement | 240 |
| heart | pacemaker or defibrillator | 287 |
| heart | pericardial effusion | 1652 |
| heart | pericardial thickening | 83 |
| heart | sternotomy | 799 |
| heart | transplant | 145 |
| lung | air trapping | 1986 |
| lung | airspace disease | 9130 |
| lung | aspiration | 877 |
| lung | atelectasis | 7031 |
| lung | bronchial wall thickening | 2873 |
| lung | bronchiectasis | 2896 |
| lung | bronchiolectasis | 1957 |
| lung | bronchiolitis | 2778 |
| lung | bronchitis | 2142 |
| lung | cancer | 2003 |
| lung | cavitation | 239 |
| lung | consolidation | 4692 |
| lung | emphysema | 4732 |
| lung | fibrosis | 3646 |
| lung | groundglass | 7978 |
| lung | hemothorax | 25 |
| lung | honeycombing | 186 |
| lung | infection | 9009 |
| lung | infiltrate | 9191 |
| lung | interstitial lung disease | 5358 |
| lung | lung resection | 213 |
| lung | mass | 1992 |
| lung | mucous plugging | 427 |
| lung | nodule | 11976 |
| lung | nodule>1cm | 1191 |
| lung | opacity | 14998 |
| lung | pleural effusion | 2844 |
| lung | pleural thickening | 1258 |
| lung | pneumonia | 8309 |
| lung | pneumonitis | 7866 |
| lung | pneumothorax | 127 |
| lung | postsurgical | 467 |
| lung | pulmonary edema | 1041 |
| lung | reticulation | 2147 |
| lung | scattered nodules/nodes | 9696 |
| lung | septal thickening | 2680 |
| lung | tree in bud | 1135 |
| lung | tuberculosis | 1490 |

