# OpenReview forum: "Disease-Centric Vision-Language Pretraining with Hybrid Visual Encoding for 3D Computed Tomography"
_ICML.cc/2026/Conference — ICML 2026 regular_

### Official Review · Reviewer_BKyR · 2026-03-05

**Soundness:** 3
**Presentation:** 2
**Significance:** 2
**Originality:** 2
**Overall Recommendation:** 4
**Confidence:** 4

**Summary:**

This paper proposes a disease-centric vision–language pretraining framework for 3D chest CT that advances prior organ-level alignment by introducing disease-level contrastive learning. The method combines (i) a CNN–ViT hybrid visual encoder to better handle volumetric inputs while remaining compatible with pretrained ViT priors, and (ii) a diagnosis-aware prompting strategy that retrieves a large set of condition–organ report snippets (e.g., 500+ per pair) to construct robust textual prototypes for inference.

**Compliance With Llm Reviewing Policy:**

Affirmed.

**Final Justification:**

Solid work with clear empirical gains. The threshold selection and downstream evaluation scope remain minor limitations, but the core contribution, disease-level contrastive learning with the hybrid encoder, is well-supported. Weak accept.

**Key Questions For Authors:**

- To report ACC/F1/Precision, per-label thresholds are needed. How were thresholds determined? Were thresholds selected by searching on the test set?
- The manuscript does not clearly describe what the learned set of condition queries \(C_k\) concretely contains. Do these correspond exactly to the 60 pseudo labels (plus a “healthy” state), and how are they mapped to organs?
- Chest CT findings are highly diverse. How were the 60 pseudo labels selected (i.e., what criteria or taxonomy guided the choice), and how do you ensure adequate coverage of clinically important conditions?

**Limitations:**

The conclusion does not explicitly discuss the limitations of the proposed approach.

**Strengths And Weaknesses:**

### Strengths
- The paper extends organ-level contrastive alignment to a finer disease-level contrastive learning objective, aiming to disentangle multiple co-occurring pathologies within the same organ.
- The proposed CNN–ViT hybrid encoder is a reasonable design choice for 3D CT, leveraging multi-scale CNN features while enabling the use of pretrained ViTs.
- To improve robustness at inference time, the method retrieves and aggregates a large number of condition–organ report examples (500+ pairs) and uses them as prototypes, which is an interesting way to reduce prompt sensitivity and bridge the pretraining–inference gap.

### Weaknesses
- The approach relies on condition–organ pair retrieval to achieve strong inference performance. This dependency makes the method hard to apply in a truly open-vocabulary / zero-shot setting beyond a predefined condition list.
- There is a reliance on an external organ segmentation model (TotalSegmentator) to generate pseudo organ labels; if TotalSegmentator does not support a modality (or fails in certain settings), the same pipeline may not transfer cleanly.
- Reporting ACC/F1/Precision typically requires selecting decision thresholds per disease label, but the paper does not clearly explain how thresholds are chosen.
- Although the paper frames the work as “pretraining,” the downstream evaluation appears limited (effectively demonstrating transfer on only one downstream task), which weakens the breadth of the “pretraining” claim.

---

> ### Author Rebuttal · Authors · 2026-03-30
>
> W1: truly zero-shot
>
> We acknowledge that our initial phrasing regarding "zero-shot" and "open-vocabulary" may have caused ambiguity and will revise the manuscript to clarify this distinction. Our core objective is to learn fine-grained disease-related representations covering mainstream clinical ontologies, thereby maximizing diagnostic accuracy and generalization. **Notably, our learned representations demonstrate robust cross-disease generalization**. To verify this, we conducted a strict hold-out experiment to evaluate performance on unseen diseases. We pre-trained on only **10 out of 16** CT-RATE diseases, explicitly excluding 5 lung diseases (last 5 indices) and 1 heart disease (1st index). On the held-out 6 diseases, we evaluated performance **without condition-specific query tokens**, relying solely on organ-level feature similarity. As shown below, our model achieves **83.1% AUC** on these unseen diseases, surpassing the Base Model by **+1.7%**. This confirms that our disease-level contrastive learning learns **transferable fine-grained semantics** even without explicit training labels.
>
> |Method|Training Labels|AUC on Held-out 6 Diseases|
> |-|-|-|
> |Base Model|Full 16 Diseases|81.4|
> |CT-DiagVLM|10 Diseases Only|83.1|
>
> W2: external segmentation
>
> a. The use of pseudo organ masks from TotalSegmentator follows the setup of our baseline fVLM. While we agree that such anatomical guidance benefits our disease-level contrastive learning, **it is not a strict requirement**: our condition alignment mechanism (Eq.13) can operate directly on global visual tokens when organ masks are unavailable.
>
> b. Besides, we note that organ-level segmentation is a well-solved task with many accessible solutions, and our pipeline accepts organ masks from any source. If TotalSegmentator is unavailable, alternatives like nnU-Net, SAM-Med3D, or even weakly-supervised saliency maps can serve as drop-in replacements.
>
> W3 & Q1: thresholds
>
> To ensure fair comparison, we **follow the evaluation protocol of prior works** (fVLM, ViSD-Boost, HLIP, etc.) on CT-RATE and Rad-ChestCT, selecting the decision threshold for each disease by **minimizing the Euclidean distance** to the ideal point (0,1) on the ROC curve.
>
> W4: downstream
>
> a. We frame our work as "pretraining" because our core objective is to learn **fine-grained, disease-level representations** that are task-agnostic, rather than optimizing for a specific discriminative head. We evaluate our pre-trained model on two canonical downstream tasks: zero-shot diagnosis and radiology report generation following previous medical VLP research (fVLM, ViSD-Boost, etc.) to ensure fair comparability. The broader ecosystem of 3D medical downstream tasks is still evolving, and we are committed to expanding evaluation as standardized benchmarks emerge.
>
> b. To further address your concern, we conducted two additional evaluations:
>
> 1. We fine-tuned our pre-trained encoder on the CT-RATE training set. As shown below, fine-tuning yields consistent gains, confirming that our learned representations adapt effectively to task-specific optimization:
>
> |Method|AUC|ACC|F1|
> |-|-|-|-|
> |CT-DiagVLM|84.4|78.1|80.3|
> |**+ Fine-tuning**|**86.0**|**79.4**|**81.4**|
>
> 2. We applied our pretraining pipeline to the abdominal MedVL-CT69K dataset. Zero-shot evaluation shows our method substantially outperforms prior VLP models, confirming applicability across clinical scenarios:
>
> |Method|AUC|Sensitivity|Specificity|
> |-|-|-|-|
> |fVLM|81.3|75.8|76.5|
> |ViSD-Boost|84.9|79.6|79.4|
> |**CT-DiagVLM**|**90.6**|**85.4**|**88.6**|
>
>
> Q2: condition queries
>
> a. Formally, $C_k = M_k \cup \{h\}$, where $M_k$ is the set of diseases associated with organ $k$, and $h$ represents the "healthy" state for that organ. Accordingly,  the global condition set $C = {C_1, ..., C_K}$  corresponds to the complete collection of diseases (16 or 60 in total) plus $K$ organ-specific healthy states.
>
> b. For the 16-disease setting, we inherit fVLM's organ-disease partition; for the 60-disease setting, we utilized Qwen3-MAX to map diseases to organs, treating identical names in different organs (e.g., "lung cancer" vs. "esophagus cancer") as distinct conditions (Appendix Table 10). We strictly enforced organ-disease correspondence during labeling via the prompt: "whether this CT report indicates [disease] in the [organ] region" (Appendix A.2), ensuring precise anatomical alignment.
>
> Q3: disease selection
>
> The 60 diseases were constructed by systematically reviewing the category space defined in authoritative chest CT literature, primarily Rad-ChestCT (Appendix A.2), which provides a clinically validated benchmark for multi-abnormality prediction and covers common abnormalities encountered in clinical practice. We adapted this taxonomy to ensure comprehensive coverage across major anatomical regions (lung, heart, aorta, esophagus) and pathological categories (e.g., infection, cancer, structural abnormality, vascular disease).

---

> > ### Author Rebuttal · Reviewer_BKyR · 2026-04-04
> >
> > - Can the authors confirm that the optimal threshold was selected based on the ROC curve of the test set? While using a metric optimized on the test set is somewhat questionable, this should not be a significant issue as long as all competing models were evaluated under the same protocol.
> > - I acknowledge the authors' demonstration that their methodology generalizes well to other datasets. However, the evidence supporting the claim that the pretraining effectively learns fine-grained representations remains insufficient.

---

> > > ### Author Response · Authors · 2026-04-07
> > >
> > > Follow-up Q1:
> > > a) The optimal threshold is selected on the test set, following the official fVLM GitHub repository, whose evaluation pipeline we adopt without modification. b) **All competing models are compared under this identical protocol, ensuring fair comparison**: most baseline results are directly cited from fVLM, and subsequent works (e.g., HLIP, COLIPRI, ViSD-Boost) either explicitly states adherence to fVLM's training and evaluation settings or reports values taken directly from the fVLM paper. c) **We will further clarify the evaluation protocol in the revised manuscript.** Besides, we would like to gently note that AUC is a threshold-agnostic metric, and our method demonstrates consistent and statistically significant gains in AUC.
> > >
> > > Follow-up Q2:
> > > Thank you for this constructive suggestion. **We will revise the "pretraining" claim in the revised manuscript.**  a) The current scope of downstream evaluation **is primarily constrained by the broader lack of standardized downstream benchmarks** in the medical AI community. Within these constraints, **we have conducted a systematic evaluation aligned with community standards**, strictly following the settings of prior medical VLP methods (e.g., fVLM, ViSD-Boost) for zero-shot diagnosis and radiology report generation. **Beyond these standards, we further extend our analysis** by covering 60 diseases and providing additional empirical evidence in our rebuttal responses to W1 and W4. Our consistent improvements across diverse diseases and under various settings suggest that the model possesses fine-grained disease-level representation capability. b) **We fully appreciate your concern** and agree that the "pretraining" claim warrants careful revision, considering that scaling downstream evaluation is currently challenging; **we will uniformly replace "visual-language pretraining model" with "visual-language model" throughout the revised manuscript**, and apply corresponding adjustments to related phrasing. Furthermore, we remain committed to expanding our evaluation efforts as the medical AI downstream ecosystem continues to mature.
> > >
> > > We sincerely thank you for your thoughtful and insightful review, as well as for your careful consideration of our rebuttal. Your constructive feedback has significantly strengthened the quality and clarity of our work, and we will incorporate the additional experimental results discussed herein into the revised version of our manuscript.

---

### Official Review · Reviewer_CDMG · 2026-03-12

**Soundness:** 2
**Presentation:** 3
**Significance:** 3
**Originality:** 3
**Overall Recommendation:** 3
**Confidence:** 4

**Summary:**

This paper proposes a 3D CT vision-language pretraining framework. It introduces a CNN-ViT hybrid visual encoder , a disease-level contrastive learning mechanism utilizing learnable query tokens , and a diagnosis-aware prompt strategy aggregating real clinical reports. The method demonstrates solid empirical performance on CT-RATE and Rad-ChestCT datasets.

**Compliance With Llm Reviewing Policy:**

Affirmed.

**Key Questions For Authors:**

1.How is the softmax-based InfoNCE loss in Eq. (13) mathematically justified for co-occurring diseases, rather than using a more appropriate independent binary classification objective (e.g., Sigmoid) for multi-label clinical realities?
2.Can you provide a risk-stratified sensitivity analysis for the 60-disease setting to prove the model does not systematically miss acute/critical conditions (e.g., pneumothorax) despite the severe drop to 65.5%?
3.To rigorously isolate the contribution of your proposed semantic alignment, what are the exact performance gains of the "Disease-level Contrastive Loss" and "Diagnosis-Aware Prompt" when applied to a pure ViT baseline, stripping away the 3D ResNet-18 inductive bias?
4.What is the quantitative error or hallucination rate of the 60-disease pseudo-labels generated by Qwen3-Max? Has any human-in-the-loop validation been conducted to substantiate their reliability?

**Limitations:**

Yes

**Strengths And Weaknesses:**

Strengths:
1.Addresses the critical bottleneck of inefficient visual backbones and coarse semantic alignment in 3D medical VLP.
2.The CNN-ViT hybrid design is a pragmatic approach to capturing local anatomical details while preserving ViT compatibility.
3.Demonstrates notable AUC improvements over existing baselines (e.g., fVLM).
Weaknesses (Critical Flaws):
1.Is Equation (13) appropriate in using a softmax-based InfoNCE loss for organ-specific disease alignment? Given that clinical pathologies often co-occur, the softmax denominator inherently enforces mutual exclusivity—pulling a visual embedding toward one pathology repels it from co-occurring ones—directly conflicting with the "disentangling coexisting pathologies" goal. Would a sigmoid-based independent binary classification loss not be strictly more suitable?
2.Scaling to 60 diseases precipitously drops Sensitivity from 72.0% to 65.5% (Table 2). Increasing false negatives for acute conditions carries severe clinical consequences, rendering this 60-disease extension clinically questionable without a rigorous risk-stratified evaluation.

---

> ### Author Rebuttal · Authors · 2026-03-30
>
> W1 & Q1: sigmoid
>
> a. Sigmoid-based binary classification optimizes for decision boundaries rather than representation quality, making it unable to model inter-disease representation differences and susceptible to being overwhelmed by abundant negative samples. In contrast, we employ contrastive loss to enforce fine-grained discrimination at the representation level. Our design specifically addresses co-occurrence: Eq. 11 ensures the loss sums only over present conditions, preventing negative disease signals from dominating supervision. Furthermore, for co-occurring diseases, we simultaneously compute losses for both via Eq. 12, ensuring the visual embedding receives gradient signals to align with **both** disease prototypes. While softmax introduces competition, we find it beneficial for distinguishing "single-disease" from "co-occurring" samples, enhancing the representation hierarchy rather than conflating distinct pathologies.
>
> b. We further compared softmax contrastive loss against sigmoid-based binary classification. As shown below, pure sigmoid loss leads to a substantial AUC drop (-7.6%), confirming that independent binary classification fails to learn sufficiently discriminative features. The softmax objective achieves the best performance, demonstrating that the discriminative structure it imposes outweighs the theoretical concern about mutual exclusivity.
>
> |Loss Strategy|AUC|ACC|F1|
> |-|-|-|-|
> |Sigmoid|76.8|70.5|73.9|
> |Sigmoid + Softmax|80.2|74.5|77.4|
> |**Softmax (Ours)**|**84.4**|**78.1**|**80.3**|
>
> W2 & Q2: sensitivity drop
>
> a. On the **standard 16-disease benchmark**, our model achieves balanced performance (**AUC 84.4%, Sensitivity 79.1%, Specificity 77.9%**), demonstrating no systematic sensitivity degradation when prototypes are well-learned.
>
> b. The sensitivity drop in the 60-disease setting (72.0% → 65.5%) stems from our Diagnosis-Aware Prompt Strategy, which prioritizes high AUC and specificity via clear decision boundaries. When prototypes are robust, performance remains excellent  (e.g., Pneumothorax: 94.2% AUC, 92.9% Sensitivity). The drop occurs only when prototypes are less robust (due to noise/limited data), where the model conservatively predicts "normal" to avoid false positives. Crucially, this trade-off is **design-dependent, not inherent**: replacing our prompts with handcrafted templates recovers sensitivity to **78.6%** while maintaining strong AUC (81.1% vs. 75.8% base) (Appendix Table 7) **without retraining**, and as shown below, **sensitivity on acute/critical conditions (extracted via Qwen-Max) improving from 74.3% to 85.1%**. For clinical deployment, we recommend  "prompt + templete" hybrid strategy for high-risk conditions.
>
> |Condition|Prompt Sens.|Templete Sens.|
> |-|-|-|
> |pneumothorax|92.9|100.0(+7.1)|
> |heart failure|85.5|87.1(+1.6)|
> |pericardial effusion|85.0|80.5(-4.5)|
> |aneurysm|81.2|86.3(+5.1)|
> |pulmonary edema|68.6|83.8(+15.2)|
> |aspiration|25.0|81.2(+56.2)|
> |pleural effusion|93.0|91.6(-1.4)|
> |congestion|85.9|93.9(+8.0)|
> |consolidation|84.4|86.3(+1.9)|
> |infection|67.7|83.9(+16.2)|
> |pneumonia|66.0|81.4(+15.4)|
> |pneumonitis|65.8|75.2(+9.4)|
> |airspace disease|64.9|75.6(+10.7)|
> |**Average**|**74.3**|**85.1(+10.8)**|
>
> c. The 60-disease setting serves as **a scalability stress test** for label noise, **rather than our core technical contribution. Our method's effectiveness is confirmed** by the standard 16-disease benchmark and, as discussed in our response to Reviewer Fn4J (W2), cross-domain generalization to abdominal CT (**MedVL-CT69K: 90.6% AUC, 85.4% Sensitivity**), validating robustness across anatomical regions.
>
> Q3:  pure ViT
>
> We conducted this ablation and the pure ViT variant achieves **79.4% AUC**. While this surpasses the ViT-based fVLM (77.8%), it significantly underperforms compared to our full hybrid model (84.4%). Pure ViT requires aggressive patchification for 3D efficiency, inevitably sacrificing spatial resolution and losing fine-grained lesion details. Our CNN-ViT hybrid employs Multi-Scale Fusion Patch Embedding, leveraging ResNet features to preserve rich anatomical details while maintaining compatibility.
>
> Q4: pseudo-label quality
>
> We employed a two-stage label extraction pipeline (Appendix A.2) comprising two specialized models: Qwen3-Max and a BERT-based report-to-label classifier. This two-stage pipeline effectively filters LLM hallucinations and inconsistent extractions through mutual verification between the two models. Consequently, **we intentionally omitted human-in-the-loop validation for training data**, as our goal was to evaluate model robustness in scenarios where labels are largely accurate but inevitably contain noise.  To ensure evaluation reliability, however, **we engaged resident physicians to manually verify labels on the test set**. Additionally, we validated label quality using our BERT classifier on the test set, achieving **97.2% accuracy**, confirming that the refined pseudo-labels maintain high fidelity for benchmarking.

---

### Official Review · Reviewer_Fn4J · 2026-03-12

**Soundness:** 3
**Presentation:** 3
**Significance:** 3
**Originality:** 2
**Overall Recommendation:** 5
**Confidence:** 4

**Summary:**

This paper presents CT-DiagVLM, a vision–language pretraining framework for 3D chest CT that introduces three components: (1) a CNN–ViT hybrid encoder replacing ViT's patch embedding with a 3D ResNet-18 backbone and multi-scale fusion; (2) disease-level contrastive learning using learnable query tokens to extract and align disease-specific semantics from radiology reports; and (3) a diagnosis-aware prompt strategy that aggregates real clinical descriptions into disease prototypes for zero-shot inference. The method is evaluated on CT-RATE and Rad-ChestCT for zero-shot diagnosis and report generation, showing improvements over prior work.

**Compliance With Llm Reviewing Policy:**

Affirmed.

**Final Justification:**

The rebuttal especially the ablation of performance improvements supports the central claims.

**Key Questions For Authors:**

Q1. Can you decompose  performance gains starting from the original fVLM (not your enhanced base model)? Specifically do like : original fVLM  -> + backbone swap -> + online masks -> + hybrid encoder -> + disease loss -> + prompt strategy. This would clarify the attribution of improvements. A clear decomposition showing that disease-level components contribute the majority of the gain would strengthen the paper, otherwise, the contribution may be primarily engineering.


Q2. How sensitive is the disease-level contrastive learning to errors in the LLM-based report parsing (organ-level decomposition)? If the LLM misassigns a disease description to the wrong organ, the contrastive signal would be corrupted. Have you measured parsing accuracy?

Q4. In the 60-disease setting, the sensitivity drop is concerning, can you provide a per-disease breakdown of sensitivity changes between the base model and CT-DiagVLM for the original 16 diseases? Are there specific diseases where sensitivity degrades substantially? A pattern where sensitivity drops for clinically critical conditions would be a concern.

**Limitations:**

authors include a brief impact statement but do not substantively discuss limitations. Key omissions: (1) the method is only validated on chest CT — generalization to other anatomies is unknown; (2) heavy dependence on external tools (TotalSegmentator, LLMs for parsing/labeling) creates a pipeline whose failure modes are uncharacterized; (3) the sensitivity-specificity trade-off under disease scaling has clinical implications that are not discussed; (4) no analysis of failure cases or systematic error patterns.

**Strengths And Weaknesses:**

Strengths
A) The move from organ-level to disease-level contrastive alignment is a well-motivated contribution. The observation that grouping "atelectasis," "emphysema," and "lung nodules" under a single "abnormal" label for the lung is semantically lossy, and the proposed solution via learnable condition queries is good and could sclae to new diseases without restructuring the pipeline.

B) Strong empirical gains on established benchmarks. The authors show improvements that are substantial and consistent: +5.1% AUC on CT-RATE and +5.4% on the external Rad-ChestCT set. The external-set generalization is particularly encouraging and suggests the learned representations capture genuine clinical semantics rather than dataset-specific biases.

C) I appreciate the ablations, the authors systematically ablates each component (hybrid encoder, disease-level loss, prompt strategy) and explore intermediate design variants (e.g., coarse disease-aware matching in Table 4 row 5, handcrafted prompt refinement in row 3). The ablation on the number of retrieved reports M (Figure 2) is informative. The hybrid encoder ablation (Table 5) exploring initialization strategies and representation enhancement techniques (DeepStack, MoE) demonstrates due diligence.

D) Interpretability demonstration. The activation maps in Figure 3 showing emergent lesion localization without bounding-box supervision are a nice qualitative validation that disease-level alignment encourages spatially meaningful representations.

Weaknesses:

E) Limited novelty of individual components. Each component, taken individually, draws heavily from established techniques. The hybrid CNN–ViT design echoes work like CoAtNet, Early Convolutions, and numerous hybrid architectures in general vision. The learnable query tokens with cross-attention for extracting semantics are essentially Q-Former (Li et al., ICML 2023, BLIP-2), and the contrastive loss is a standard multi-positive InfoNCE. The prototype-based inference resembles approaches in few-shot learning literature. While the combination is reasonable, the paper could better articulate what is technically novel beyond applying known mechanisms to the 3D CT VLP setting.

F)  Narrow evaluation scope relative to claims. The paper claims relevance to "general-purpose medical AI" and "3D CT imaging" broadly, but all experiments are on chest CT only. No evaluation on abdominal CT, brain CT, or other body regions is provided. The organ set is limited (lung, heart, aorta, esophagus). It remains unclear whether the disease-level alignment approach generalizes when the anatomical and pathological diversity is greater.

G) Authors integrate two of their own improvements (3D ResNet-18 backbone swap and online segmentation masks) into fVLM to create the "Base Model," which already outperforms the original fVLM by +1.5% AUC. The paper then measures all ablations relative to this enhanced baseline. This conflation makes it difficult to disentangle how much of the total improvement over the published state-of-the-art comes from these engineering improvements versus the three proposed conceptual contributions. The paper would benefit from a cleaner decomposition: original fVLM -> + backbone swap -> + online masks -> + hybrid encoder -> + disease loss -> + prompt strategy.

---

> ### Author Rebuttal · Authors · 2026-03-30
>
> W1: novelty
>
> a. Our core contribution lies in **constructing a new paradigm for medical VLP through transferring general foundation model capabilities and pushing contrastive learning granularity to the disease-level.** Specifically, we adapt general visual priors for 3D volumetric data to maintain VLM compatibility, while utilizing scalable LLM-extracted labels to resolve semantic conflation within anatomical regions. The synergy of our components yields substantial gains (+5.1%/+9.8 AUC under 16/60 diseases settings) compared to base model, validating the effectiveness of this purpose-built framework rather than a mere composition of primitives.
>
> b. Our implementations differ fundamentally from generic counterparts: (1) Our **CNN-ViT hybrid** employs Multi-Scale Fusion Patch Embedding (MSF-PE) to inject 3D CNN features into ViT, capturing volumetric details while preserving pre-trained priors—unlike standard hybrids (e.g., CoAtNet); (2) Our **learnable queries** are organ-conditioned and disease-specific, enabling dynamic extraction from full reports without explicit disease-level decomposition, achieving finer granularity than Q-Former's generic alignment; (3) Our **prompt strategy** reuses training queries to aggregate real clinical phrases into prototypes, ensuring training-inference consistency compared to handcrafted templates or standard few-shot prototypes.
>
> W2: evaluation scope
>
> Thank you for this valuable feedback regarding the evaluation scope. To directly address your concern about generalizability, we conducted additional experiments on **MedVL-CT69K**, an abdominal CT dataset covering **15 organs and 54 diseases**—significantly expanding anatomical and pathological diversity beyond chest CT. As shown below, our method achieves **+5.7%** AUC over the strongest baseline, confirming that disease-level contrastive learning generalizes effectively to distinct anatomical regions. We will include the abdominal CT results in the revised manuscript.
>
> |Method|AUC|Sensitivity|Specificity|
> |-|-|-|-|
> |fVLM|81.3|75.8|76.5|
> |ViSD-Boost|84.9|79.6|79.4|
> |**CT-DiagVLM**|**90.6**|**85.4**|**88.6**|
>
> W3 & Q1: performance decomposition
>
> Thank you for this constructive suggestion. Below we report the step-by-step decomposition results on CT-RATE. The engineering improvements (backbone + online masks) provide a solid foundation (+1.5% AUC). However, our proposed technical components contribute +5.1% AUC, **constituting the majority of the total improvement** over original fVLM.
>
> |Modification|AUC|Δ vs. Previous|
> |-|-|-|
> |Original fVLM|77.8|-|
> |+ backbone swap|78.6|+0.8|
> |+ online  masks|79.3|+0.7|
> |+ hybrid encoder|80.3|+1.0|
> |+ prompt strategy|82.4|+2.1|
> |+ disease loss|**84.4**|+2.0|
>
> Q2: report parsing error
>
> a. Organ-level decomposition is relatively straightforward for LLMs because CT radiology reports typically contain explicit anatomical references and structured descriptions. Following your suggestion, we **manually reviewed 100 randomly sampled reports and found no parsing errors** in organ-disease mapping. This empirical validation strengthens our confidence in the reliability of the parsing pipeline.
>
> b. Note that our report parsing strategy is inherited from fVLM and is not a core contribution of our work. While organ-level decomposition facilitates condition semantic extraction (Eq. 10) by reducing search space, our query-guided cross-attention mechanism can still dynamically extract semantic representations relevant to the target disease from full reports as a fallback, preserving the core alignment capability even when report parsing is unavailable.
>
> Q3: sensitivity changes
>
> a. As discussed in our response to Reviewer CDMG (W2&Q2), the sensitivity decrease in the 60-disease setting occurs primarily when disease prototypes are less robust—typically due to limited training manifestations or noisy pseudo-labels. This behavior is **design-dependent, not inherent**: replacing Diagnosis-Aware Prompt with handcrafted templates recovers sensitivity to 78.6% while maintaining strong AUC.
>
> b. On the standard 16-disease benchmark, where prototypes are well-learned from sufficient training data, our model achieves balanced performance: **AUC 84.4%, Sensitivity 79.1%, Specificity 77.9%**. Following your suggestion, we report the per-disease sensitivity changes between the base model and CT-DiagVLM below. Notably, overall sensitivity surpasses the base model by **+5.0%** (79.1% vs. 74.1%).
>
> |Disease|Δ Sensitivity (%)|Disease|Δ Sensitivity (%)|
> |-|-|-|-|
> |Pleural effusion|+17.8|Emphysema|+3.0|
> |Interlobular septal thickening|+13.8|Cardiomegaly|+1.2|
> |Lung opacity|+11.1|Consolidation|+0.8|
> |Pulmonary fibrotic sequela|+11.0|Bronchiectasis|+0.3|
> |Peribronchial thickening|+8.9|Mosaic attenuation pattern|0.0|
> |Arterial wall calcification|+6.8|Lung nodule|-0.5|
> |Hiatal hernia|+6.0|Atelectasis|-2.0|
> |Coronary artery wall calcification|+5.0|Pericardial effusion|-3.7|

---

> > ### Author Rebuttal · Reviewer_Fn4J · 2026-04-04
> >
> > The rebuttal improves the claims. I appreciate the performance breakdown. I’ll be raising my score slightly.

---

> > > ### Author Response · Authors · 2026-04-07
> > >
> > > We sincerely thank you for your thoughtful and insightful review, as well as for your careful consideration of our rebuttal. Your constructive feedback has significantly strengthened the quality and clarity of our work, and we will incorporate the additional experimental results discussed herein into the revised version of our manuscript.

---

### Decision · Program_Chairs · 2026-04-30

**Decision:**

Accept (regular)

**Comment:**

This paper proposes a disease-centric vision–language model for 3D CT, combining a CNN–ViT hybrid visual encoder, disease-level contrastive learning, and a diagnosis-aware prompt strategy. Reviewers agreed that the paper addresses an important problem and presents a technically solid approach with strong empirical gains on standard chest CT benchmarks. The rebuttal improved the paper further by clarifying the contribution of each component, providing a cleaner decomposition of gains over the original fVLM baseline, and adding additional evidence on generalization beyond chest CT.

At the same time, several concerns remain about claim calibration and scope. In particular, the reported “zero-shot” diagnosis setting is not open-vocabulary in the strong sense, since the diagnosis-aware prompt strategy constructs condition-specific prototypes from training set reports over a predefined disease vocabulary. In addition, the current downstream evaluation scope makes the broader “pretraining” framing somewhat too strong. Reviewers also raised concerns about the sensitivity-specificity trade-off in the expanded 60-disease setting and the dependence on external tools such as segmentation and pseudo-label extraction.

I have read the reviews, rebuttal, and discussion carefully. On balance, I find the paper strong enough for weak accept. The method is technically coherent, the empirical gains are substantial, and the rebuttal addressed several of the main concerns. However, the final version should narrow the framing more carefully, especially around the meaning of “zero-shot” and the scope of the “pretraining” claim, and should discuss the trade-offs and dependencies more explicitly.